# Elucidating the Functions of Non-Coding RNAs from the Perspective of RNA Modifications

**DOI:** 10.3390/ncrna7020031

**Published:** 2021-05-11

**Authors:** Venkata Naga Srikanth Garikipati, Shizuka Uchida

**Affiliations:** 1Department of Emergency Medicine, The Ohio State University Wexner Medical Center, Columbus, OH 43210, USA; venkata.garikipati@osumc.edu; 2Dorothy M. Davis Heart Lung and Research Institute, The Ohio State University Wexner Medical Center, Columbus, OH 43210, USA; 3Center for RNA Medicine, Department of Clinical Medicine, Aalborg University, Frederikskaj 10B, 2. (building C), DK-2450 Copenhagen SV, Denmark

**Keywords:** epitranscriptomics, non-coding RNA, RNA modifications

## Abstract

It is now commonly accepted that most of the mammalian genome is transcribed as RNA, yet less than 2% of such RNA encode for proteins. A majority of transcribed RNA exists as non-protein-coding RNAs (ncRNAs) with various functions. Because of the lack of sequence homologies among most ncRNAs species, it is difficult to infer the potential functions of ncRNAs by examining sequence patterns, such as catalytic domains, as in the case of proteins. Added to the existing complexity of predicting the functions of the ever-growing number of ncRNAs, increasing evidence suggests that various enzymes modify ncRNAs (e.g., ADARs, METTL3, and METTL14), which has opened up a new field of study called epitranscriptomics. Here, we examine the current status of ncRNA research from the perspective of epitranscriptomics.

## 1. Introduction

Just as DNA and proteins, RNA can be modified by a variety of enzymes. There are 170+ RNA modifications known to date [1,2], which is much higher than DNA and proteins. Historically, RNA modifications have been studied for those of housekeeping RNAs, such as ribosomal RNAs (rRNAs) and transfer RNAs (tRNAs) [3,4,5,6,7] and only recently that the field of RNA modifications (called epitranscriptomics [8]) has been extended to other RNA molecules, including protein-coding genes, microRNAs (miRNAs) and long non-coding RNAs (lncRNAs). This expansion of the epitranscriptomic field is the result of the recent development of high-throughput methods, such as mass spectrometry and next-generation sequencing (NGS), especially RNA sequencing (RNA-seq) [9,10,11,12,13]. Among many RNA modifications, the most famous one in recent years is N6-methyladenosine (m^6^A), which is reversible methylation of nitrogen-6 position of adenosine (A) [14,15]. This modification frequently occurs in most eukaryotes, especially in the messenger RNAs (mRNAs) [16,17,18,19] and has been shown to increase during organ maturation and development [20,21,22,23] especially in the adult brain [24], cancer pathogenesis, and progression [25,26]. Because of increased interest in epitrascriptomics, there is an explosion in the number of publications covering various aspects of cellular processes, organ developments, and disease progressions for many different RNA modifications, including m^6^A, RNA editing, and pseudouridylation. These RNA modifications have contributed significantly to RNA metabolism, including transcription, splicing, subcellular location, stability, and translation. Thus, in this review, we will summarize the current status of epitranscriptomics by focusing on the biogenesis of lncRNAs and their functions to provide a concise view of epitranscriptomics in lncRNA biology.

## 2. Different Types of RNA Modification Marks Reported for lncRNAs

Like DNA and histones, RNA undergoes epigenetic modifications termed epitranscriptomics, which exploded over the recent years studying 170+ RNA modifications [1,2]. In this review article, we focus on the following RNA modifications: m^6^A, N1-methyladenosine (m^1^A), adenosine (A) to inosine (I) RNA editing, 5-methylcytidine (m^5^C), and pseudouridine (Ψ) [27] (Figure 1).

### 2.1. m^6^A Modification

The methylation of adenosine is catalyzed by two writer proteins, methyl transferase 3 and methyl transferase 14 (METTL3 and METTL14), forming m^6^A methyl transferase complex (MTC) [28,29]. Moreover, Wilms tumor 1-associated protein (WTAP) acts as an adaptor protein that recruits other RNAs and proteins to MTC to target multiple RNA transcripts [30,31,32]. Recent studies identified other adaptor proteins, such as RNA-binding motif protein 15 (RBM15). RBM15B interacts with MTC in a WTAP-dependent manner and represses the lncRNA X-inactive specific transcript (*XIST*) transcription [33].

Recent studies uncovered methyl transferase 16 (METTL16) as another writer protein in the MTC. Importantly, crosslinking sites of METTL16 were found in the ACm^6^AGAGA motif, which is mainly found in intronic regions. Intriguingly, ACm^6^AGAGA motif was associated with spliceosomes [34,35], suggesting subsets of m^6^A methyltransferases have diverse functions, and future studies focusing on understanding the role of METTL16 in alternative splicing is warranted. On the other hand, recent studies underscored demethylases acting as eraser proteins. Two such proteins, fat mass- and obesity-associated protein (FTO) and alkylation repair homolog protein 5 (ALKBH5), gained much attention recently, emphasizing the concept that m^6^A is a reversible and dynamic process [36]. Mechanistically, FTO oxidizes m^6^A to unstable intermediates N6-hydroxymethyladenosine (hm^6^A) or N6-formyladenosine (f^6^A), which further hydrolyzes to adenine. In contrast, ALKBH5 removes the adenosine’s methyl group directly [37]. A group of RNA binding proteins (YTH N6-methyladenosine RNA binding proteins:**** YTHDF1, YTHDF2, YTHDF3, and YTHDC1) specifically recognize the methylated adenosine on RNA that participates in RNA stability or translation (reviewed in [38]). Overall, m^6^A writers, erasers, and readers participate in a complex mechanism crucial for lncRNA functions, especially related to pathogenesis and progression of various diseases, such as those summarized in the following paragraphs (Table 1).

Just as mRNAs, m^6^A modification stabilizes lncRNAs to affect their functions. Examples include metastasis-associated lung adenocarcinoma transcript 1 (*MALAT1*); one of the most studied lncRNA that is involved in many disease conditions [39]. Interestingly, *MALAT1* owns multiple m^6^A sites (A2515, A2577, A2611, and A2720) [40]. The *MALAT1* hairpin contains the domain for the binding of heterogeneous nuclear ribonucleoprotein C (HNRNPC) and m^6^A methylation site (A2577) [40]. In the presence of m^6^A mark, methylation destabilizes hairpin and enhances HNRNPC binding. Recent work demonstrated that YTHDC1, an m^6^A reader, recognizes m^6^A marks on *MALAT1*, which plays a crucial role in maintaining the expression of vital oncogenes via reshaping nuclear spots and genomic binding [41]. Simultaneously, m^6^A-deficient *MALAT1* rescued the metastatic nature of esophageal cancer cells [41], suggesting the functional role of m^6^A marks on *MALAT1*.

As in the case of *MALAT1*, most of the published studies on the effects of m^6^A marks on the functions of lncRNAs are related to cancers. For example, a study investigating lncRNA expression profile in 502 head and neck squamous cell carcinoma (HNSCC) patients identified a significantly elevated level of lncRNA activating regulator of DKK1 (*LNCAROD*) associated with tumor stage and reduced overall survival [42]. Mechanistically, the m^6^A modification through m^6^A writers, METTL3 and METTL14, stabilizes *LNCAROD* expression. Then, *LNCAROD* by forming a ternary complex with heat shock protein family A (Hsp70) member 1A (HSPA1A) and Y-box binding protein 1 (YBX1) promotes HNSCC disease progression [42]. In pancreatic cancer, the m^6^A demethylase.

**Table 1 ncrna-07-00031-t001:** A non-exhaustive list of m^6^A modified lncRNAs.

LncRNA	m^6^A Regulator	Function	References
*MALAT1*	YTHDC1	Reshapes the composition of nuclear spots and enhances oncogene expression.	[41]
*LNCAROD*	METTL3 and METTL14	Promotes HNSCC disease progression.	[42]
*KCNK15-AS1*	ALKBH5	Enhances pancreatic cell migration and invasion.	[43]
*DANCR*	IGF2BP2	Enhances pancreatic cancer cell growth and tumorigenesis.	[44]
*PVT1*	ALKBH5	Promotes osteosarcoma cell proliferation, migration, and invasion.	[45]
*GAS5*	YTHDF3	Involved in colorectal cancer.	[46]
*RP11-138 J23.1*	METTL3	Progresses colorectal cancer.	[47]
*FAM225A*	METTl3	Enhances nasopharyngeal carcinoma cell, proliferation, migration, invasion, and metastasis.	[48]
*LINC00958*	METTl3	Promotes hepatocellular carcinoma.	[49]
*linc1281*	METTl3	Induces mouse embryonic stem cell differentiation.	[50]
*BDNF-AS*	METTL3	Increases the risk of AUD.	[51]
*Olfr29-ps1*	METTl3	Promotes immunosuppressive function and differentiation of myeloid-derived suppressor cells.	[52]

(Eraser) ALKBH5 inhibits the disease progression by demethylating the lncRNA KCNK15 and WISP2 antisense RNA 1 (*KCNK15-AS1*) and increasing the stability of *KCNK15-AS1*, while the down-regulation of *KCNK15-AS1* inhibited the cell migration and invasion [43]. Besides m^6^A writers and erasers, m^6^A readers also affect the functions of lncRNAs via m^6^A marks. For example, the m^6^A reader, insulin-like growth factor 2 mRNA binding protein 2 (IGF2BP2), regulates the stability of differentiation antagonizing non-protein coding RNA (*DANCR*), which is crucial for pancreatic cancer cell growth and tumorigenesis [44].

Yes1 associated transcriptional regulator (YAP) has been a critical factor in colorectal cancer (CRC) progression [53]. Interestingly, a functional link between m^6^A modification and lncRNA in YAP signaling and CRC was reported. In this study [46], an m^6^A reader, YTHDF3, is a novel target of YAP that can promote the degradation of m^6^A modified lncRNA, growth arrest-specific 5 (*GAS5*), in CRC progression. In regards to m^6^A modified lncRNAs, another lncRNA was reported to be involved in CRC. In this study [47], the lncRNA *RP11-138 J23.1* (*RP11*) was elevated in CRC patients with the disease progression. *RP11* promoted CRC cell proliferation and metastasis ability by suppressing the proteasomal degradation of the transcription factor, zinc finger E-box binding homeobox 1 (ZEB1); thereby enhancing epithelial to mesenchymal transition. m^6^A RNA-immunoprecipitation (RIP) assays revealed an increased association of *RP11* to m^6^A antibody in CRC cells compared to control cells. Further, the overexpression of METTL3 increased *RP11* expression in the CRC cells, suggesting m^6^A-induced *RP11* expression promotes CRC progression and is likely a potential biomarker and a novel therapeutic target for CRC [47]. The m^6^A modifications on lncRNAs can also play a vital role in their binding efficiency to miRNAs and/or proteins. For example; the significance of lncRNAs in nasopharyngeal carcinogenesis (NPC) was addressed in a recent study [48]. Using a microarray-based screening approach, the authors identified a lncRNA, family with sequence similarity 225 member A (*FAM225A*), as the most upregulated lncRNA in NPC associated with poor survival in these patients. *FAM225A* enhanced cancer cell proliferation, migration, invasion, and metastasis by sponging *miR-590-3p* and *miR-1275*, which activate integrin-β3 and PI3K/AKT cell survival pathways. The authors identified two RRACU m^6^A consensus motifs in the last exon of *FAM225A*. Subsequent methylated RNA immunoprecipitation (Me-RIP) assay revealed elevated m^6^A levels in NPC cell lines (SUNE-1 and HONE-1) compared to control nasopharyngeal epithelial cell lines (NP69 and N2Tert). Further, METTL3 knockdown experiments revealed 50% to 60% reduced total FAM225A expression, suggesting the role of m^6^A marks on the *FAM225A* stability in NPC patients [48]. Another well characterized oncogenic lncRNA is Pvt1 oncogene (*PVT1*) [54]. A study shows that the expression of *PVT1* was elevated in osteocarcinoma (OS) tissues and significantly associated with clinical stage, tumor size, and prognosis of OS patients [45]. Mechanistically, m^6^A demethylase (eraser), ALKBH5, interacts with *PVT1* to prevent its degradation. Long non-coding RNA 00958 (*LINC00958*) has been shown to be upregulated in gastric, glioma, gynecological, oral, and pancreatic cancer [49]. However, its role in hepatocellular carcinoma (HCC) was unclear. To address this lack of information, a study demonstrated that METTL3-mediated m^6^A modification enhanced the stability of *LINC00958* [49]. Mechanistically, *LINC00958* acts as miRNA sponge against *miR-3619-5p* to increase the expression of its target gene, hepatoma-derived growth factor (HDGF), which promotes HCC growth [49]. The functional connection between m^6^A marks and lncRNAs as miRNA sponges has been shown in several studies. One such study demonstrated that m^6^A modification of a rodent-specific lncRNA, *linc1281*, acting as miRNA sponge against *let-7*, induced mouse embryonic stem cell differentiation [50]. At the same time, mutations or deletion of METTL3 abolished *linc1281* binding to *let-7*. Another study identified a lncRNA pseudogene, olfactory receptor 29, pseudogene 1 (*Olfr29-ps1*), was upregulated in myeloid-derived suppressor cells (MDSCs) upon proinflammatory stress induced by interleukin 6 (IL-6) [52]. Mechanistically, *Olfr29-ps1* sequestered *miR-214* to promote immunosuppressive function and differentiation of MDSCs, whereas these effects were abolished with silencing METTL3 [52]. The above studies highlight that m^6^A modifications in lncRNAs dictate their binding efficiency and functions.

Besides their role in various cancers, it is now clear that m^6^A modifications in lncRNA involvement in alcohol use disorders (AUD). For instance, it is well known that adolescent alcohol drinking contributes to developing AUDs in adulthood. In this perspective, an elegant study was performed to understand a link between m^6^A and lncRNAs in the postmortem amygdala of adolescent drinking individuals (subjects who started drinking alcohol before or at 21 years of age) [51]. The authors observed that reduced m^6^A modification in the lncRNA BDNF antisense RNA (*BDNF-AS*) results in the increased levels of *BDNF-AS*, subsequently repressing brain-derived neurotrophic factor (BDNF) expression, a critical factor in the central nervous system and increasing the risk of AUDs, indicating m^6^A-mediated action on lncRNAs in another disorder [51].

Circular RNAs (circRNA) are covalently circularized RNA loops that are mainly generated by pre-mRNA splicing (reviewed in [55]). CircRNAs exhibits differential expression in response to cellular stress events and various human diseases, including cancers and cardiovascular and neurodegenerative diseases [55,56,57,58,59]. Owing to their unique circular structure, they are naturally inaccessible to exonucleases, resulting in their increased half-life. Thus, circRNAs represent reliable biomarkers to detect various human diseases. How circRNAs degrade within cells has been an unanswered question. In this regard, a study revealed P/MRP endonuclease cuts m^6^A containing circRNAs via recruiting its reader, YTHDF2, and its adaptor protein, reactive intermediate imine deaminase A homolog (HRSP12 or RIDA), to degrade circRNAs [60]. Although circRNAs are considered non-coding, except for a few circRNAs that exhibited peptide/protein-coding ability [61,62], a remarkable study demonstrated that single nucleotide m^6^A modification on circRNAs is sufficient to promote translation via the recruitment of initiation factor initiation factor eukaryotic translation factor 4 gamma (elF4G2) and m^6^A reader, YTHDF [63]. A significant proportion of circRNAs come from open reading frame (ORF) containing protein-coding genes; however, what drives ORF-circRNA biogenesis was unclear. In this context, a study revealed that m^6^A modification in circRNA drives the biogenesis of circRNA with coding potential in mouse male germ cells [64]. Taken together, future studies are warranted to address the functions of the circRNA-derived peptides and proteins.

The role of m^6^A modifications on circRNAs in immune responses has been a focus of recent research efforts. A study demonstrated how mammalian cells detect foreign circRNAs and induce innate immunity [65]. In this study, the m^6^A modifications in endogenous or self circRNAs inhibit innate immunity, while the absence of m^6^A modifications in exogenous or foreign circRNAs activates RNA pattern recognition regulation of RIG-I (retinoic acid-inducible gene I; official gene name, DExD/H-box helicase 58 (DDX58)) to induce a robust immune response by activating T-cells and antibody production in human cells [65]. In contrast, another study showed that the transfection of purified circRNAs did not induce an immune response and, therefore, propose circRNA potential therapeutic usage without adverse immune reactions [66]. Taken together, the role of circRNAs in the immune-stimulatory function needs further investigation, especially addressing how endogenous circRNAs achieve self-tolerance and how m^6^A or other RNA modifications on circRNAs contribute to inflammation in endogenous vs. foreign circRNAs.

In summary, with the ongoing research on lncRNAs and m^6^A modification, it has become evident that m^6^A modifications regulate multiple biological functions, especially in various cancers. Future research needs to address the role of m^6^A modifications on lncRNAs in other human diseases. Furthermore, m^6^A modifications on circRNA sequences and their functional relevance in health and disease need to be thoroughly explored.

### 2.2. m^1^A Modification

m^1^A forms by introducing a methyl group to the N1 position of adenosine. It is not as abundant as m^6^A. Unlike m^6^A, m^1^A writers include TRMT6, TRMT61A, TRMT10C; readers for m^1^A sites include YTHDF1, YTHDF2, YTHDF3, and YTHDC1, which are mainly similar to m^6^A. An exciting aspect of m^1^A is its erasers, ALKBH3 and ALKBH1, which ensure demethylation of m^1^A [67,68,69]. Dysregulation of m^1^A on tRNA has been reported in cancers, and an elevated level of ALKBH3 has been reported in pancreatic cancer [70]. However, how m^1^A modifications affect lncRNA structures and functions is still not evident.

### 2.3. Adenosine (A) to Inosine (I) RNA Editing

Conversion of adenosine (A) to inosine (I) by adenosine deaminase (ADAR) is another prevalent form of RNA modification [71]. Three of these gene family members were identified in vertebrates [72]. ADAR1 and ADAR2 are expressed in most tissues, whereas ADAR3 is only expressed in the brain [73]. A-to-I change results in changes in transcripts and alternative splicing. A-to-I modifications in lncRNAs exhibited cancer progression and cardiovascular disease [71,74]. Moreover, ADAR1-mediated A-to-I changes in prostate cancer antigen 3 (*PCA3*) lncRNA increased its binding with PRUNE2 pre-mRNA to promote cancer cell proliferation, migration, and invasion [75]. Using hoc indexing and de novo editing events, a comprehensive inosinome in lncRNAs was performed in the healthy brain cortex and glioblastoma [76]. The authors identified >10,000 new sites and 335 novel lncRNAs that undergo editing, suggesting the A-to-I RNA editing on lncRNAs maintains the physiology of healthy brain as well as its dysregulation is linked to tumor progression [76].

### 2.4. m^5^C Modification

m^5^C is a methylated form of cytosine (C), which is well studied in DNA, tRNA, and rRNA. Enzymes NOL1/NOP2/SUN domain family member (NSUN) family, NSUN1 to NSUN7, and DNA methyltransferase-like 2 (DNMT2) have been reported to participate in this RNA modification [77]. A recent study using modified RNA bisulfite sequencing identified m^5^C sites on lncRNAs in HeLa cells, with a low stoichiometry [78]. An elegant study has shown that methylated cytosine sites in the functional domain of the lncRNA *HOTAIR* and *XIST* are essential for binding to chromatin-associated protein complexes [79]. Furthermore, another study identified comprehensive methylated cytosine in the epitranscriptome of the mouse brain and embryonic stem cells in ncRNAs as well, although low in number [80]. These findings highlight the role of cytosine methylation modifications on lncRNAs. This is encouraging, and future studies on m^5^C modification on lncRNAs would help us better understand their biological and functional role in health and disease.

### 2.5. Ψ Modificatio

Ψ is known as the “fifth nucleotide” due to its abundance and represents the most prevalent RNA modifications. Pseudouridine is an isomer of nucleoside uridine catalyzed by Ψ synthase (PUS) that removes nitrogen-carbon glycosidic bond and replaces its carbon-carbon glycosidic bond [81]. The presence of 170 Ψ sites was recently identified in lncRNAs; interestingly, in well-characterized lncRNAs, including *MALAT1*, *XIST*, and KCNQ1 opposite strand/antisense transcript 1 (*KCNQ1OT1*) [82,83]. Thus, future studies are warranted to uncover functional and mechanistic insights of Ψ in lncRNAs in the context of health and disease.

## 3. Impact of Epitranscriptomic Marks on lncRNA Structures

When the first draft of the human genome was introduced, there were high hopes for understanding many of nature’s rules about the human body. Two decades later, we have realized that there is more to human genes than simply looking at DNA sequences. The same situation applies to elucidating the functions of lncRNAs. Many researchers were excited to read about terminal differentiation-induced non-coding RNA (*TINCR*) as the authors identified TINCR box motifs, which are 25-nt long RNA sequences that interact with many other mRNAs [84]. The discovery of TINCR box motifs prompted a further search for similar binding domains of other lncRNAs. However, such screening did not yield fruitful results [85,86]. Not only was such search not successful, but it also recently became clear that *TINCR* owns an evolutionary conserved open reading frame, which encodes for peptides of 87 amino acids [87]. Within this TINCR peptide, one of 10 TINCR box motifs is included, suggesting that sequence alone cannot be used to infer functions of lncRNAs. There are a number of methods proposed and used to predict the functions of lncRNAs by combining different features of lncRNAs, including evolutionary-conserved sequence motifs, secondary structures, and potential binding of RNA-binding proteins and miRNAs [85,86,88,89]. Yet, none of such methods can predict the functions of all lncRNAs, which is not surprising as not all protein-coding genes have been functionally characterized. In addition to the current challenges facing the computational functional predictions of lncRNAs, growing evidence of epitranscriptomic marks on lncRNAs is of particular interest as yet another parameter that researchers need to consider when investigating the functions of lncRNAs and other types of RNA species.

More than half of the human genome is made up of repetitive sequences [90]. The Ensembl database currently classifies these repeat sequences into 10 classes (centromere, low complexity regions, RNA repeats, satellite repeats, simple repeats, tandem repeats, LTRs (long tandem repeats), SINE (short interspersed nuclear element), LINE (long interspersed nuclear element), and Type II transposons) and categorize those that cannot be classified into above 10 classes as “Unknown” (https://m.ensembl.org/info/genome/genebuild/assembly_repeats.html accessed on 22 March 2021). Not surprisingly, such repetitive sequences are also present in lncRNAs [91,92,93]. For example, the subfamily of SINE, Alu elements, can be found in 11% of the human genome [94]. These 300-nt repetitive repeats are derived from transposons and exist only in primates. These elements can be expressed as their own RNA [95] or parts of other transcripts (e.g., introns of mRNAs, lncRNAs), where their expression levels increase upon stresses (e.g., heat shock, hypoxia, viral infection) [96,97]. When two Alu elements in opposite directions meet, they form double-stranded RNA, which can be recognized by RNA-binding proteins, such as ADARs. The ADAR-mediated A-to-I changes also occur frequently in lncRNAs [98,99,100]. Not surprisingly, these A-to-I conversions change the secondary structures of RNA [101], which is also an important point to be considered when analyzing for lncRNA functions as the binding of other macromolecules (i.e., DNA, RNA, and proteins) can alter depending on the presence (or absence) of double-stranded RNA motifs within a lncRNA [102,103,104].

Besides A-to-I RNA editing, other epitranscriptomic marks affect the structures of lncRNAs. In particular, m^6^A marks are of interest as it has been shown to be in a negative relationship with A-to-I RNA editing [105]. More recently study shows that silencing of the m^6^A writer, METTL3, in glioma stem-like cells altered A-to-I and C-to-U RNA editing (another type of RNA editing, which is less frequent than A-to-I) events by differentially regulating RNA editing enzymes ADAR and APOBEC3A, respectively [106]. An interesting model is proposed recently regarding m^6^A marks affecting the secondary structure of one of the most well studied lncRNA, *MALAT1* [107]. By performing secondary data analyses of dimethyl sulfate-sequencing (DMS-Seq) data from human erythroleukemic cell line K562 and psoralen analysis of RNA interactions and structure (PARIS) data from cervical cancer-derived HeLa cells compared to the working structural model of *MALAT1* in noncancerous cells, the authors postulated that m^6^A-based structural changes of *MALAT1* might mediate cancer in a cell-type-specific manner [107]. Thus, increasing evidence suggests that examining epitranscriptomic marks on lncRNAs is important to uncover the potential functions of lncRNAs [108].

## 4. Secondary Analysis of RNA-Seq and m^6^A-Seq Data to Reveal the Impact of m^6^A Marks on lncRNAs

As stated earlier, the most well-studied epitranscriptomic marks in recent years are m^6^A. Indeed, a number of lncRNAs have been shown to own m^6^A marks, including *lincRNA 1281* (official gene name; ephemeron, early developmental lncRNA (*Eprn*)) during the differentiation of mouse embryonic stem cells [50], *MALAT1* in obstructive nephropathy [109], *Pvt1* in sustaining stemness of epidermal progenitor cells [110], and *XIST* in transcriptional repression [33]. Many more m^6^A-marked lncRNAs are found in cancers, such as FOXF1 adjacent non-coding developmental regulatory RNA (*FENDDR*) in endometrioid endometrial carcinoma [111], KCNK15 and WISP2 antisense RNA 1 (*KCNK15-AS1*) in pancreatic cancer [43], nuclear paraspeckle assembly transcript 1 (*NEAT1*) in the colon [112] and gastric cancer [113], as well as miRNA sponges, including long intergenic non-protein coding RNA 857 (*LINC00857*) in pancreatic cancer [114], long intergenic non-protein coding RNA 958 (*LINC00958*) in breast cancer [115], and *PVT1* in osteosarcoma [45]. Furthermore, there are several high-throughput screening studies reporting m^6^A marks in lncRNAs [16,24,116,117,118,119] as well as databases (CVm^6^A [120], DRUM [121], m^6^A-Atlas [122], m^6^Acorr [123], M6A2Target [124], m^6^AVar [125], MeT-DB [126,127], REPIC [128], RMBase [129,130], RMDisease [131], RMVar [132], and RNAWRE [133]) have been released with m^6^A marks (and also other epigenetic marks, such as m^1^A, m^5^C, RNA editing, and Ψ, for some databases) in protein-coding and lncRNAs. These screenings are the results of RNA immunoprecipitation (RIP) using anti-m^6^A antibody (often termed as m^6^A-seq) [18] as well as more elaborate techniques, such as amplicon sequencing evaluation method for RNA m^6^A sites after chemical deamination (NOseq) [134], MeRIP-seq (methylated RNA immunoprecipitation sequencing) [24], and miCLIP (m^6^A individual-nucleotide-resolution cross-linking and immunoprecipitation) [118] technologies. As there are several enzymes involved in m^6^A (writers, readers, and erasers), in some screening studies, knockdown of each m^6^A enzyme was performed to record the dependency of m^6^A marks for each enzyme. Yet, most of these studies mainly focus on protein-coding genes as further biological validation experiments are possible by examining protein expressions of m^6^A-marked transcripts that encode. Furthermore, it is not clear whether m^6^A-dependent (based on m^6^A marks) and -independent effects of m^6^A enzymes (changes in gene expressions due to the loss of a particular m^6^A enzyme but not affecting m^6^A marks) in such screening studies.

We performed a secondary analysis of previously published RNA-seq data to address the above point directly. In the original study [135], the authors report that METTL14 is a crucial component for the crosstalk between histone H3 trimethylation at Lys36 (H3K36me3, a marker for pre-mRNA splicing) and m^6^A marks. Using the human hepatoma cell line HepG2, RNA-seq and m^6^A-seq experiments were performed upon silencing of m^6^A writers—*METTL3*, *METTL14*, and *WTAP* (Gene Expression Omnibus (GEO) accession GSE110320). Since the original study analyzed only for protein-coding genes and used the older version of the genome, hg19, a secondary analysis of this data set was performed using the latest annotation provided by the Ensembl database (GRCh38.103). The usage of the latest annotation of the human genome is crucial as the number of lncRNAs has been increased drastically in recent years, which allows us to examine the lncRNAs more carefully. Of 19,796 protein-coding and 16,593 lncRNA genes without readthrough transcripts registered under the GRCh38.103 annotation file, there are less than 100 differentially expressed genes (both protein-coding and lncRNA genes) upon silencing of each m^6^A writer compared to the control (HepG2 cells treated with short hairpin RNA (shRNA) against non-specific sequences) (Figure 2A,B, Appendix A. When up-and down-regulated protein-coding genes are compared among silencing of m^6^A writers, only two protein-coding genes (*HMGCS2* (3-hydroxy-3-methylglutaryl-CoA synthase 2) and *LGALS2* (galectin 2)) are shared between silencing of *METTL3* and *METTL14*, while three protein-coding genes (*BHMT* (betaine--homocysteine S-methyltransferase), *CYP4F2* (cytochrome P450 family 4 subfamily F member 2), and *STMN4* (stathmin 4)) are shared between silencing of *METTL3* and *WTAP*. However, no protein-coding gene is shared among all three conditions (Figure 2C). In the case of down-regulated protein-coding genes, no gene was shared (Figure 2D). Unlike the protein-coding genes, only 11 non-overlapping lncRNA genes are identified as up-regulated (Figure 2E). Simultaneously, there is no down-regulated lncRNA gene in the silencing of any of three m^6^A writers compared to the control, suggesting that at the level of transcriptional control, m^6^A writers influence different sets of protein-coding genes. However, such transcriptional influence is rather minimal, especially since such transcriptional control via m^6^A writers is negligible for lncRNA genes, even for the mRNA-seq (targeting only RNA with poly A tails; thus missing about half of lncRNAs without poly A tails).

## 5. Materials and Methods

### 5.1. RNA-Seq Data Analysis

RNA-seq data were downloaded from the Sequence Read Archive (SRA) database using SRA Toolkit. [136] FASTQ files were preprocessed with fastp [137] (version 0.21.0) using the default setting. After preprocessing of sequencing reads, STAR [138] (version 020201) was used to map the reads to the reference genome (GRCh38.103). To calculate counts per million (CPM) values and derive differentially expressed genes, the R package, edgeR [139] (version 3.30.3), was used. False discovery rate (FDR)-adjusted *p*-values were used for further analysis.

### 5.2. Data Analysis and Visualization

Volcano plots were generated using R-package, ggplot2 [140]. To draw heat maps, MultiExperiment Viewer (MeV) [141] was used.

## 6. Conclusions

On the whole, we summarize updates on lncRNA epitranscriptomics, in the context of lncRNA function and biology. Even though the last couple of decades of research revealed the importance of epitranscriptomics in health and disease, several questions still need to be answered, such as future insights into the functional importance of RNA modification in lncRNAs? Are these modifications conserved between species, whether these modifications are mediators or actual drivers? How do we identify different modifications on the same lncRNA? Can these modifications be targeted to restrict disease progression? All these above questions would unravel our understanding of epitranscriptomics as novel disease mechanisms to design effective and targeted therapeutics.

## Figures and Tables

**Figure 1 ncrna-07-00031-f001:**
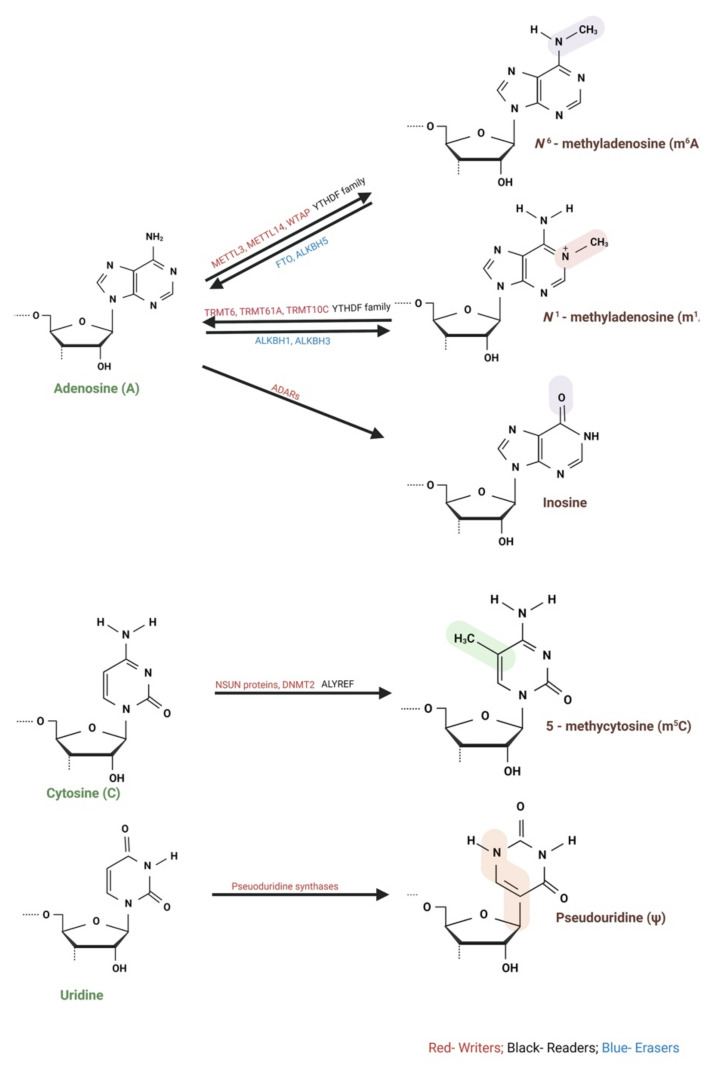
Schematic representations of RNA modifications. ADAR, adenosine (A) to inosine (I) RNA editing; AlyREF, Aly/REF export factor; ALKBH proteins, alkB homolog; DNMT2, DNA methyltransferase-like 2; FTO, fat mass and obesity-associated protein; m^1^A, N1-methyladenosine; m^6^A, N6-methyladenosine; m^5^C, 5-methylcytosine; METTL, methyltransferase-like; NSUN, NOP2/Sun domain family members; PUS, pseudouridine synthase; RNMT, RNA guanine-7 methyltransferase; RPUSD, RNA pseudouridine synthase domain-containing protein; TRM6, transfer RNA methyltransferase non-catalytic subunit 6; TRM61, transfer RNA methyltransferase catalytic subunit 61; TRMT10, transfer RNA methyltransferase 10; YTHDC, YTH domain-containing; YTHDF, YTH domain-containing family. Created with BioRender.com accessed on 31 March 2021.

**Figure 2 ncrna-07-00031-f002:**
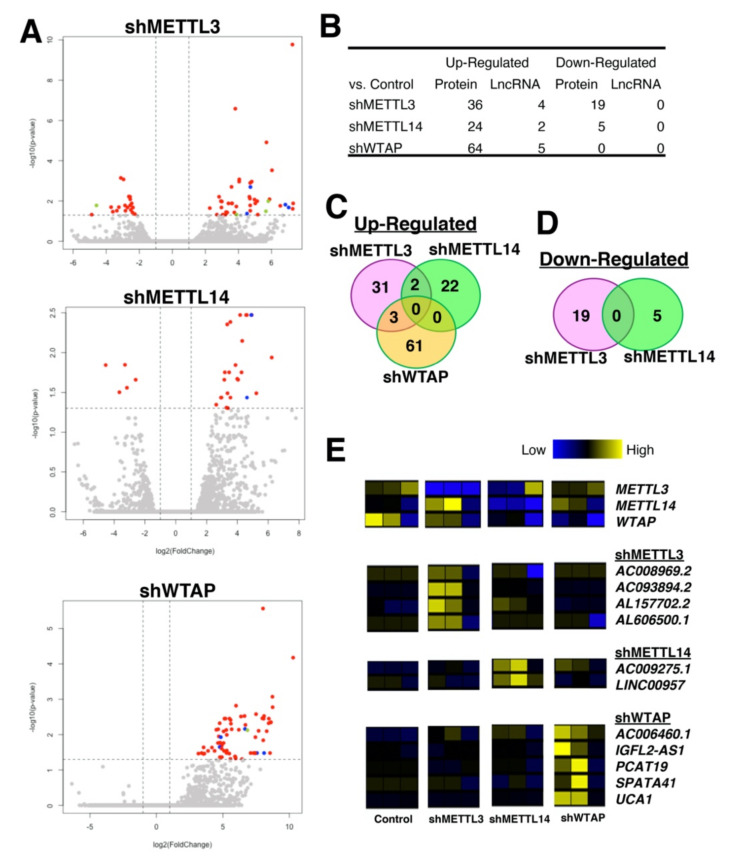
RNA-seq data of silencing of m^6^A writers in HepG2 cells. (**A**) volcano plots comparing silencing of *METTL3*, *METTL14*, or *WTAP* to the control sample group. With the threshold values of 2-fold and FDR-adjusted *p* < 0.05, protein-coding genes are colored in red, lncRNA genes in blue, and other genes (e.g., pseudogenes) in yellow-green. (**B**) the table indicates the number of differentially expressed genes for each category. (**C**,**D**) Venn diagrams for shared protein-coding genes that are (**C**) up- and (**D**) down-regulated in each condition. (**E**) heatmaps of target m^6^A reader genes followed by up-regulated lncRNA genes in each condition.

## Data Availability

The commands and programs used in this study can be found in the Github repository (https://github.com/heartlncrna/Analysis_of_GSE110320). The data sets analyzed in this study can be found in the Zenodo repository (https://doi.org/10.5281/zenodo.4635589, accessed on 22 March 2021).

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
