# Peer review of "Elucidating the Functions of Non-Coding RNAs from the Perspective of RNA Modifications"

_ncrna, 2021, doi:10.3390/ncrna7020031_

Round 1
Reviewer 1 Report
In this review manuscript, Garikipati and Uchida summarized the types of RNA modifications of non-coding RNAs and their functions. It’s interesting to focus on the RNA modifications of non-coding RNAs and elucidating their functions. I have some comments.
Major comments:
- The authors introduced several RNA modifications and the modifications on non-coding RNAs, which is not very in depth. The authors should summarize in more details in this section, current manuscript is not very clear and the authors didn’t provide what the true function of these RNA modifications are and how the RNA modifications work. It would also be great if the authors could provide a table showing the different RNA modifications and their role/functions and how.
- The authors spent a lot of efforts on a secondary analysis of RNA-seq and MeRIP-seq data and showed a minimum effect of knockdown of m^6A writers on both coding and long-non-coding RNAs. In my view, this analysis didn’t provide too much information as it’s well known that m^6A mediates the RNA translation, RNA decay, RNA stability and translocation and so on instead of impairing its cognate mRNA expression. I would suggest to narrow down this section.
Minor comments:
- Besides m^6A, m^1A, A-to-I editing, pseudouridine (ψ) and m^5C, there are some other RNA modifications found in non-coding RNAs. It’s better to mention them.
- Besides the m6A modification on mRNA and lncRNA, there are many literatures described m^6A modifications in other non-coding RNAs, for example, circRNA.
- In the manuscript, m6A should be m^6A, the ‘6’ should be a superscript, so do the others, m^1A, m^C and so on.
- Page 7 line #259, ‘readers’ should be ‘writers’.
Author Response
In this review manuscript, Garikipati and Uchida summarized the types of RNA modifications of non-coding RNAs and their functions. It’s interesting to focus on the RNA modifications of non-coding RNAs and elucidating their functions. I have some comments.
Response: Thank you very much for your valuable comments. We have revised our manuscript to reflect your comments below:
Major comments:
The authors introduced several RNA modifications and the modifications on non-coding RNAs, which is not very in depth. The authors should summarize in more details in this section, current manuscript is not very clear and the authors didn’t provide what the true function of these RNA modifications are and how the RNA modifications work. It would also be great if the authors could provide a table showing the different RNA modifications and their role/functions and how.
Response: We have revised our manuscript extensively to address the above point by focusing more on human diseases, especially for m6A modifications, as there are many studies published in this regard.
The authors spent a lot of efforts on a secondary analysis of RNA-seq and MeRIP-seq data and showed a minimum effect of knockdown of m^6A writers on both coding and long-non-coding RNAs. In my view, this analysis didn’t provide too much information as it’s well known that m^6A mediates the RNA translation, RNA decay, RNA stability and translocation and so on instead of impairing its cognate mRNA expression. I would suggest to narrow down this section.
Response: We have narrow down this section and removed Figure 3 about m6A-seq.
Minor comments:
Besides m^6A, m^1A, A-to-I editing, pseudouridine (ψ) and m^5C, there are some other RNA modifications found in non-coding RNAs. It’s better to mention them.
Response: Since there are over 170 RNA modifications known to date, we cannot cover all modifications in this manuscript. Thus, we focused on those modifications that have been studied recently in relation to lncRNAs. To highlight this point, the following sentence was added to Section 2:
"Like DNA and histones, RNA undergoes epigenetic modifications termed epitran-scriptomics, which exploded over the recent years studying 170+ RNA modifications [1,2]. In this review article, we focus on the following RNA modifications: m6A, N1-methyladenosine (m1A), Adenosine (A) to inosine (I) editing, 5-methylcytidine (m5C), and pseudouridine (Ψ) [27] (Figure 1)."
Besides the m6A modification on mRNA and lncRNA, there are many literatures described m^6A modifications in other non-coding RNAs, for example, circRNA.
Response: Thank you very much for pointing out this important point. We have now added two paragraphs about circRNAs and their relation to m6A modifications.
In the manuscript, m6A should be m^6A, the ‘6’ should be a superscript, so do the others, m^1A, m^C and so on.
Response: The above points have been corrected accordingly.
Page 7 line #259, ‘readers’ should be ‘writers’.
Response: The above points have been corrected accordingly.
Reviewer 2 Report
The manuscript by Garikipati and Uchida reviews in its first half RNA modifications and their potential influences on long non-coding (lnc)RNAs functions and structure. In its second half, the authors re-analysed some previously published data, RNA-seq and m6A-seq, to determine the effects of knocking down different m6A writers, METTL3, METTL14, and WTAP, on lncRNAs expression and on m6A peaks distribution on lncRNAs.
The main issue is that the manuscript is not really a review, as the first half is short and provides only a limited overview of RNA modifications found on lncRNAs, with hardly any description of the effects of RNA modifications on lncRNAs functions (likely due to the fact that these effects are still mostly unknown). The second part of the manuscript does not really add anything. Furthermore, the data re-analyzed have been performed with an oligo (dT) purification step, which will affect the analysis of lncRNAs as they are known to be poorly polyadenylated (for lncRNAs, data have to be from a total RNA-seq with ribodepletion, not poly(A)+ selected).
Other comments:
- Line 80: typo in N6-hydroxymethyladenosine.
- Line 110: HELA -> HeLa
- Line 220: H3K36me3 is a marker of pre-mRNA splicing, not transcriptional elongation (intronless and histone genes are transcribed but have no/extremely low level of H3K36me3).
- Line 313 & 318: Typos in epitranscriptomics
- Figure 3: It would have been more interesting to compare the m6A peaks -/+ knockdown to see how many peaks are gained/lost/unchanged after m6A writers’ removal and what are the overlaps between the different knockdowns. Currently, Figure 3C does not provide any meaningful information.
Author Response
The manuscript by Garikipati and Uchida reviews in its first half RNA modifications and their potential influences on long non-coding (lnc)RNAs functions and structure. In its second half, the authors re-analysed some previously published data, RNA-seq and m6A-seq, to determine the effects of knocking down different m6A writers, METTL3, METTL14, and WTAP, on lncRNAs expression and on m6A peaks distribution on lncRNAs.
The main issue is that the manuscript is not really a review, as the first half is short and provides only a limited overview of RNA modifications found on lncRNAs, with hardly any description of the effects of RNA modifications on lncRNAs functions (likely due to the fact that these effects are still mostly unknown). The second part of the manuscript does not really add anything. Furthermore, the data re-analyzed have been performed with an oligo (dT) purification step, which will affect the analysis of lncRNAs as they are known to be poorly polyadenylated (for lncRNAs, data have to be from a total RNA-seq with ribodepletion, not poly(A)+ selected).
Response: Thank you very much for your valuable comments. We have substantially modified our manuscript by adding more information about each epitranscriptomic marks and reducing the parts about secondary analysis of previously published RNA-seq data.
Other comments:
Line 80: typo in N6-hydroxymethyladenosine.
Response: The above points have been corrected accordingly.
Line 110: HELA -> HeLa
Response: The above points have been corrected accordingly.
Line 220: H3K36me3 is a marker of pre-mRNA splicing, not transcriptional elongation (intronless and histone genes are transcribed but have no/extremely low level of H3K36me3).
Response: The above points have been corrected accordingly.
Line 313 & 318: Typos in epitranscriptomics
Response: The above points have been corrected accordingly.
Figure 3: It would have been more interesting to compare the m6A peaks -/+ knockdown to see how many peaks are gained/lost/unchanged after m6A writers’ removal and what are the overlaps between the different knockdowns. Currently, Figure 3C does not provide any meaningful information.
Response: We have narrow down this section and removed Figure 3 about m6A-seq.
Round 2
Reviewer 1 Report
I have no more questions.
Reviewer 2 Report
The authors answered my comments.